# Sacsin Deletion Induces Aggregation of Glial Intermediate Filaments

**DOI:** 10.3390/cells11020299

**Published:** 2022-01-16

**Authors:** Fernanda Murtinheira, Mafalda Migueis, Ricardo Letra-Vilela, Mickael Diallo, Andrea Quezada, Cláudia A. Valente, Abel Oliva, Carmen Rodriguez, Vanesa Martin, Federico Herrera

**Affiliations:** 1Biosystems & Integrative Sciences Institute, Faculdade de Ciências, Universidade de Lisboa, 1649-004 Lisbon, Portugal; fernanda.murtinheira@gmail.com (F.M.); mafalda.migueis1998@gmail.com (M.M.); ricardo.letra.vilela@gmail.com (R.L.-V.); mickael_diallo@hotmail.com (M.D.); andreagtzq@gmail.com (A.Q.); 2Departamento de Química e Bioquímica, Faculdade de Ciências, Universidade de Lisboa, 1749-016 Lisbon, Portugal; 3Instituto de Tecnologia Quimica e Biologica (ITQB-NOVA), Universidade Nova de Lisboa, 2780-157 Oeiras, Portugal; oliva@itqb.unl.pt; 4Instituto de Farmacologia e Neurociências, Faculdade de Medicina, Universidade de Lisboa, 1649-028 Lisboa, Portugal; cvalentecastro@medicina.ulisboa.pt; 5Instituto de Medicina Molecular João Lobo Antunes, Faculdade de Medicina, Universidade de Lisboa, 1649-028 Lisboa, Portugal; 6Instituto Universitario de Oncología del Principado de Asturias (IUOPA), Facultad de Medicina, Universidad de Oviedo, 33006 Oviedo, Spain; carro@uniovi.es (C.R.); martinvanesa@uniovi.es (V.M.); 7Departamento de Morfología y Biología Celular, Facultad de Medicina, Universidad de Oviedo, 33006 Oviedo, Spain

**Keywords:** ARSACS, GFAP, nestin, vimentin, glioma, STAT3

## Abstract

Autosomal recessive spastic ataxia of Charlevoix-Saguenay (ARSACS) is a neurodegenerative disorder commonly diagnosed in infants and characterized by progressive cerebellar ataxia, spasticity, motor sensory neuropathy and axonal demyelination. ARSACS is caused by mutations in the SACS gene that lead to truncated or defective forms of the 520 kDa multidomain protein, sacsin. Sacsin function is exclusively studied on neuronal cells, where it regulates mitochondrial network organization and facilitates the normal polymerization of neuronal intermediate filaments (i.e., neurofilaments and vimentin). Here, we show that sacsin is also highly expressed in astrocytes, C6 rat glioma cells and N9 mouse microglia. Sacsin knockout in C6 cells (C6^Sacs−/−^) induced the accumulation of the glial intermediate filaments glial fibrillary acidic protein (GFAP), nestin and vimentin in the juxtanuclear area, and a concomitant depletion of mitochondria. C6^Sacs−/−^ cells showed impaired responses to oxidative challenges (Rotenone) and inflammatory stimuli (Interleukin-6). GFAP aggregation is also associated with other neurodegenerative conditions diagnosed in infants, such as Alexander disease or Giant Axonal Neuropathy. Our results, and the similarities between these disorders, reinforce the possible connection between ARSACS and intermediate filament-associated diseases and point to a potential role of glia in ARSACS pathology.

## 1. Introduction

The autosomal recessive spastic ataxia of Charlevoix-Saguenay (ARSACS) is a rare, early-onset neurodegenerative disorder, usually diagnosed at gait initiation (12–18 months) [1]. ARSACS is clinically characterized by cerebellar ataxia, spasticity, axonal demyelination, sensory-motor peripheral neuropathy, amyotrophy, dysarthria, skeletal finger and feet abnormalities, nystagmus, retinal hypermyelination and variable intellectual dysfunction [2]. Its histopathological features include the atrophy of the anterior vermis, associated with the loss of Purkinje cells in the cerebellum [1,2] and deposits of lipofuscin in cerebellar cortical neurons and skin [3]. ARSACS is caused by mutations in the SACS gene that are located on chromosome 13q12 and encode the cytoplasmic protein sacsin [2]. Sacsin knockout mice show histopathological and neurological features consistent with ARSACS, indicating that the disease is caused by sacsin loss-of-function [4]. Sacsin function remains barely understood, but the nature and architecture of its domains suggest that it is a molecular chaperone involved in protein quality control [5,6], mitochondrial integrity [7,8] and the assembly/dynamics of neuronal intermediate filaments [9,10].

Intermediate filaments (IFs) are one of the three major filament types of the cytoskeleton and are found in most eukaryotic cells, including neurons and glia in the central nervous system. They play a pivotal role in the mechanical and viscoelastic properties of cells and tissues, but they are also scaffolds associated with signal transduction [11]. IFs can interact with various cellular components, organelles, and molecules through linker proteins from the plakin gene family, such as plectin [12]. Unlike microtubules and actin microfilaments, IFs comprise an extremely diverse family of proteins that shows cell- and tissue-specific expression patterns [13]. For example, neuronal IF networks are constituted by vimentin and light, medium and heavy neurofilaments, while astroglial IFs are vimentin, nestin and the glial fibrillary acidic protein (GFAP). Mutations in IFs or their interacting proteins underlie more than 80 rare disorders, including Epidermolysis Bulbosa Simplex, progeria, Alexander disease (AxD), Giant Axonal Neuropathy (GAN) or predisposition to Amyotrophic Lateral Sclerosis [13].

Scientific literature on sacsin function and dysfunction was largely focused on neuronal cells, just as it happened historically for more common neurodegenerative disorders, such as Huntington’s, Parkinson’s, and Alzheimer’s diseases. However, glial cells play key roles in neurodegenerative disorders [14]. AxD is an extreme but illustrative example of the important role of glial cells in neurodegeneration. AxD is a fatal leukodystrophy caused by dominant mutations in GFAP, the major intermediate filament of astrocytes [15]. Mutant GFAP accumulates in astrocytes in cytoplasmic inclusions known as Rosenthal fibers. As ARSACS, AxD is most commonly diagnosed in the first months of life (0–2 years), and AxD patients often display ataxia, spasticity, seizures and dysarthria [15]. GFAP aggregation is also present in GAN, another early onset neurological disease characterized by an extensive aggregation of different types of Ifs [16].

In this sense, there are strong indications of a wider expression pattern for sacsin in both neural and non-neural cells and tissues, which could also be relevant for understanding both sacsin normal function and ARSACS pathology. The Protein Atlas public database indicates that medium-high sacsin mRNA levels can be found in human primary cells and cell lines from virtually any tissue, although there are significant variations between cell types within the same tissue. This database does not provide specific information on sacsin expression levels in most non-neuronal brain cells, such as astrocytes, oligodendrocytes or microglia, but it shows that Müller glia and macrophages, similar in some respects to astrocytes and microglia, respectively, express sacsin mRNA. Human glioblastoma cell lines with mixed features of astrocytes and neural precursor cells also express sacsin and, according to Harmonizome and BrainRNASeq public transcriptomics datasets, high levels of sacsin mRNA were also found in both mouse and human astrocytes. Mouse sacsin RNA expression is as high in astrocytes as in neurons, displaying the highest levels in younger animals (postnatal day 7) and decreasing with age [17]. Human fetal astrocytes express the same RNA expression levels as neurons, also decreasing as astrocytes mature [18]. In both mouse and human, neurons and astrocytes express higher levels than oligodendrocytes, microglia and endothelial cells. However, these data focus on mRNA levels, and data on protein expression are scarce.

In this report, we show that the sacsin protein is indeed expressed in astroglia. We developed an astroglial model of ARSACS by deleting sacsin in the C6 rat glioma cell line. Our results indicate that sacsin also regulates glial IF organization, suggesting a potential link between the ARSACS, AxD and GAN pathologies.

## 2. Materials and Methods

### 2.1. Reagents

Dulbecco’s modified Eagle’s medium (DMEM) was purchased from Biowest (Nuaillé, France); Fetal bovine serum (FBS), rotenone, dimethylsulfoxide (DMSO), 2′,7′-dichlorodihydrofluorescein diacetate (DCFH-DA) and 3-4,5-dimethylthiazol-2-yl-2,5-diphenyltetrazolium bromide (MTT) were acquired from Sigma-Aldrich (St. Louis, MO, USA). Penicillin-Streptomycin (Pen/Strep), dihydroethidium (DHE), 4’,6-Diamidino-2-Phenylindole, Dihydrochloride (DAPI), Pierce ECL Plus Western Blotting Substrate, MitoTracker Red CMXRos and Hoechst 33,342 were purchased from Invitrogen, Life Technologies (Carlsbad, CA, USA). Protease inhibitor cocktail was purchased from Abcam. Phosphatase inhibitor cocktail was purchased from NZYTech (Lisboa, Portugal). Mouse anti-Sacsin (N-terminal), anti-GFAP, anti-Nestin, anti-Vimentin and anti-GAPDH antibodies were purchased from Santa Cruz Biotechnology Inc. (Dallas, TX, USA). Anti-Sacsin (C-terminal) and anti-GFAP rabbit polyclonal antibodies were purchased from Millipore-Sigma™ (St. Louis, MO, USA). Anti-vimentin rabbit polyclonal antibody, horseradish peroxidase-conjugated secondary antibodies—goat anti-rabbit IgG (H + L) and goat anti-mouse IgG (H+L)—and goat anti-mouse or anti-rabbit secondary antibodies conjugated to AlexaFluor 488 or AlexaFluor 594 fluorophores were purchased from Invitrogen, Life Technologies (Carlsbad, CA, USA). Leukemia inhibitory factor (LIF) was purchased from Rockland Immunochemicals (Pottstown, PA, USA). Interleukin-6 (IL-6) and the soluble IL-6 receptor (IL-6R) were obtained from R&D Systems (Minneapolis, MN, USA). Protein extracts from the murine microglial cell line N9 were a kind gift from Drs. Adelaide Fernandes (Faculdade de Farmácia, Universidade de Lisboa). Protein extracts from a mouse embryonic frontal cortex neural precursor cell line and adult precursor cells from the rat dentate gyrus and subgranular zone were kindly provided by Drs. Susana Solá (Faculdade de Farmácia, Universidade de Lisboa, Portugal) and Sara Xapelli (Instituto de Medicina Molecular João Lobo Antunes, Lisbon, Portugal), respectively.

### 2.2. Cell Cultures and Treatments

C6 rat glioma cells, HeLa human cervix adenocarcinoma cells and HEK293T human embryonic kidney cells were acquired from ATCC (references CCL-107™, CRM-CCL-2 and CRL-1573, respectively). Human Dermal Fibroblasts (HDFn) were obtained from Innoprot (Derio, Spain) (P10857), and Human Keratinocytes (KER) from Gibco (Waltham, MA, USA) (C-100-5C). These cell lines were grown to confluence in DMEM, supplemented with 10% (*v*/*v*) FBS, 1% (*v*/*v*) Pen/Strep and maintained at 37 °C in a humidified atmosphere containing 5% CO_2_, except for KER, which were cultured in Epilife medium (Gibco^®^) (Waltham, MA, USA), 1% Pen/Strep, 1% HKGS (Gibco^®^) (Waltham, MA, USA), with 0.6 mM CaCl_2_. All cell lines were periodically tested for mycoplasma contamination by means of a commercial test (Biontex Laboratories, Munich, Germany).

For experiments, cells were plated on sterile plastic dishes or on sterile glass coverslips and allowed to adhere for 16–24 h before experiments and/or sample preparation. When applicable, C6 and C6^Sacs−/−^ cells were incubated with rotenone (5 μM) for 4 h or its vehicle (DMSO 0.1% *v*/*v*). For cytokine experiments, complete medium was removed, cells were washed with PBS and incubated in serum-free medium for 2 h before adding LIF (200 ng/mL), IL-6 (30 ng/mL) and/or the soluble IL-6 receptor (IL-6R, 60 ng/mL). Cells were then incubated for 2 h with the cytokines in the absence of serum before sample preparation.

### 2.3. Generation of Sacsin KO Cell Lines Using the CRISPR/Cas9 System

A sacsin knockout strain from C6 rat glioma cells was generated using a commercial Sacsin CRISPR/Cas9 knockout plasmid set (sc-404592, Santa Cruz Biotechnologies, Dallas, TX, USA), following manufacturers’ instructions. Transfected cells were isolated by fluorescence-activated cell sorting (FACS) and single cells were plated in 96-well plates for clonal expansion. Resulting clones were probed for sacsin expression by Western blot.

### 2.4. Primary Cultures of Astrocytes

Cultures were prepared as described elsewhere [19]. Briefly, astrocyte-enriched cultures were prepared from neonatal Sprague Dawley rat pups’ cerebral cortexes (0–2 days). Animals were sacrificed by decapitation and the brains were dissected in ice-cold PBS (NaCl 137 mM, KCl 2.7 mM, Na_2_HPO_4_.2H_2_O 8 mM and KH_2_PO_4_ 1.5 mM, pH 7.4). Cortexes were isolated, placed in 10 mL of 4.5 g/L glucose DMEM, supplemented with 10% FBS and 1% antibiotic/antimycotic, and dissociated mechanically through up-and-down movements with a serological pipette until no cell clumps were observed. Cell suspension was filtered successively through 230 μm and 70 μm (BD Falcon, NJ, United States) cell strainers and centrifuged at room temperature (RT) at 200× *g* for 10 min. The final pellet was resuspended in 4.5 g/L glucose DMEM, and cells were seeded according to the required assay. Cultures were kept at 37 °C in a humidified atmosphere (5% CO_2_) and medium was changed twice a week. At 10 days in vitro (DIV), plates were shaken for 6 h in an orbital shaker at 300 rpm to remove any contaminating microglial cells and obtain astrocytic-enriched cultures.

### 2.5. Cell Viability Assays

For MTT assays, C6 and C6^Sacs−/−^ cells were seeded onto 96-well plates at a concentration of 10^4^ cells/well. After the corresponding treatments, cells were incubated with MTT (0.5 mg/mL) for 2 h. The medium was then replaced with DMSO (100% *v*/*v*). After 15 min of incubation in the dark at RT, absorbance was determined using an automatic microplate reader (Tecan Sunrise Microplate Reader) (Tecan, Männedorf, Switzerland) at 490 nm. For LDH assays, cells were seeded in 24-well plates at a concentration of 10^5^ cells/well, and the determination of released and total LDH was carried out by means of the LDH cytotoxicity detection kit (Takara), following manufacturer’s instructions. DAPI exclusion assay was also used as a complementary cytotoxicity assay by flow cytometry, as described below.

### 2.6. Flow Cytometry

Cells were seeded in 6-well plates (5 × 10^5^ cells/well). After treatments, cells were detached by trypsin (0.05% *w/v*), collected, resuspended in PBS, and incubated with 10 μM DCFH-DA (All reactive oxygen species) or DHE (superoxide radicals) in DMEM medium (without serum) for 20 min at 37 °C in the dark. After washing cells twice with PBS, pellets were resuspended in PBS with DAPI (1 µg/mL) to discriminate between live and dead cells, and fluorescence was immediately analyzed by means of a BD LSRFortessa X-20 cell analyzer (BD Biosciences, San Jose, CA, USA). At least 10,000 events (low velocity) were recorded for analysis with FLOWJO software Version 9 (Emerald Biotech Co., Ltd., Córdoba, Spain).

### 2.7. Western Blot

Western blot analysis was performed as previously described [20]. Briefly, cells were lysed using an NP-40 lysis buffer (150 mM NaCl, 50 mM Tris-HCl pH 7.4/7.5, 1% NP-40 *v*/*v*) supplemented with protease inhibitors (NZYTech) (Lisboa, Portugal) and phosphatase inhibitors (Halt Phosphatase Inhibitor Single-use Cocktail, Thermo Fisher Scientifics, Waltham, MA, USA). Samples were sonicated in UP200s sonicator (Hielscher Ultrasonics GmbH, Teltow, Germany) for 8 s. Cell suspension was then centrifuged at 10,000× *g* for 10 min at 4 °C, and the soluble protein fraction was collected and quantified by the Bradford method, incubating 1 μL of sample with 200 μL of Bradford solution (Alfa Aesar, Ward Hill, MA, USA) for 5 min and reading the absorbance at 595 nm. Thirty micrograms of total protein was separated by SDS-PAGE on 10% (*w/v*) polyacrylamide gels (for detection of STAT3) or gradient gels 6% + 15% (*w/v*) (for detection of sacsin and IFs), and transferred to a nitrocellulose membrane. Protein transfer quality was assessed by Ponceau S staining. Membranes were blocked with 5% (*w/v*) milk in TBS-T (TBS supplemented with 0.1% Tween-20) and probed with primary antibodies in 5% (*w/v*) Bovine Serum Albumin (BSA) in PBS overnight at 4 °C. Primary antibodies were used at a dilution of 1:1000, except for anti-GAPDH and anti-sacsin antibodies (N-terminal, sc-515118, Santa Cruz Biotechnologies (Dallas, TX, USA); and C-terminal, ABN1019, Merck-Millipore (Burlington, MA, USA) which were used at 1:2000 and 1:200 dilutions, respectively. Membranes were then washed three times with TBS-T for 10 min, followed by 2 h incubation with HRP-conjugated secondary antibodies (1:10,000) in blocking solution. After washing membranes three times with TBS-T for 10 min, chemiluminescence detection was performed using the Pierce ECL Plus Western Blotting Substrate and the Amersham Imager 680 blot and gel imager (Cytiva, Marlborough, MA, USA). The integrated intensity of each band was calculated using computer-assisted densitometry analysis with ImageJ software, normalized to the loading control GAPDH as appropriate.

### 2.8. Filter Trap Assays

Cells were washed with PBS 1X and collected by scraping in native lysis buffer (173 mM NaCl, 50 mM Tris pH 7.4, 5 mM EDTA) supplemented with protease and phosphatase inhibitor cocktail. Samples were sonicated for 10 s at 5 mA using a UP200S Sonicator (Hielscher, Teltow, Germany). Protein extracts were collected after sample centrifugation at 10,000× *g* for 10 min at 4 °C and quantified by means of the Bradford method. One hundred μg of native protein extracts were diluted in PBS to produce a final volume of 100 μL and SDS was added to a final concentration of 1% (*w/v*). Samples were loaded on a dot-blotting device and filtered by vacuum through nitrocellulose membranes previously incubated with 1% (*w/v*) SDS solution in PBS. After filtration, membranes were washed twice with 1% (*w/v*) SDS solution in PBS and processed for immunoblotting detection, as described above.

### 2.9. Fluorescence Microscopy of Live Cells and Immunocytochemistry

For fluorescent microscopy, C6 and C6^Sacs−/−^ cells were seeded at the density of 10^5^ cells/cm^2^ on 35 mm glass-bottom dishes. Twenty-four hours after seeding, cells were incubated with either sirActin kit (Spirochrome, Stein am Rhein, Germany) and Tubulin tracker deep red (Invitrogen, Carlsbad, CA, USA) following manufacturer’s instructions. For visualization of intermediate filaments, cells were incubated with MitoTracker Red CMXRos (25 nM) for 30 min at 37  °C in 5% CO_2_ atmosphere before processing samples for immunocytochemistry. Cells were fixed in ice-cold methanol for 20 min and blocked with 1% (*w/v*) BSA in PBST (PBS supplemented with 0.1% Tween-20) for 1 h at RT. Overnight incubation at 4 °C was performed with the following primary antibodies diluted in blocking solution: mouse monoclonal anti-sacsin (1:50), rabbit polyclonal anti-GFAP (1:200), mouse monoclonal anti-nestin (1:100) and mouse monoclonal anti-vimentin (1:100). Cells were incubated with the corresponding goat anti-mouse or anti-rabbit secondary antibodies conjugated with AlexaFluor 488 (Thermo Fisher Scientifics, Waltham, MA, USA) or AlexaFluor 594 (Thermo Fisher Scientifics, Waltham, MA, USA) (1:800) for 2 h at RT, and counterstained with the nuclear marker Hoechst 33342 (Thermo Fisher Scientifics, Waltham, MA, USA). Images were acquired by means of a Leica TCS SPE high-resolution spectral confocal system (Wetzlar, Germany) equipped with a Leica DFC 365 FX camera (Wetzlar, Germany) using a 63X/1.4 oil objective (Wetzlar, Germany) and processed by Leica LAS X Core (Wetzlar, Germany) and ImageJ software (National Institutes of Health, Bethesda, MD, USA). The total numbers of reference C6 cells counted in three independent experiments were 549 (GFAP), 464 (Nestin) and 353 (Vimentin). The total numbers of C6^Sacs−/−^ cells counted in three independent experiments were 836 (GFAP), 697 (Nestin) and 438 (Vimentin).

### 2.10. Statistical Analysis

Statistical analysis and graphical representation of data were performed using GraphPad Prism software Version 8 (GraphPad, San Diego, CA, USA). Sample data are represented as mean ± standard error (SEM) of three independent experiments. For statistical evaluation, one-way or two-way Analysis of Variance (ANOVA) and Tukey’s post hoc test were used for multiple comparisons. Student’s *t*-test was applied for comparisons in experiments with two groups. Results were considered significant when *p* < 0.05.

## 3. Results

### 3.1. Astroglia Express Sacsin

Public databases indicated that glial cells contain sacsin mRNA, but data are scarce and there is no empirical evidence that sacsin mRNA is actually translated into protein. The sacsin protein is easily detected by immunoblotting in rat primary astroglia and C6 rat glioblastoma cells at approximately the same levels (Figure 1A). Sacsin levels in glial cells were relatively higher than in other human and rodent cell lines, some of them described to express medium-high levels of sacsin mRNA in The Protein Atlas, such as HEK293 or HeLa cells (Figure 1B,C). Surprisingly, C6 and N9 rat microglial cells had higher sacsin levels than the HT22 mouse cell line, of neuronal origin (Figure 1C). We failed in our attempts to detect sacsin in adult rat neural precursor cells from the dentate gyrus and the subventricular zone (data not shown). The immunocytochemistry of astrocytes and C6 glioma cells showed the expected cytoplasmic distribution of sacsin with some mitochondrial localization (Figure 1D). These data suggest that sacsin expression is not exclusive to neurons, but also expressed in glial cells.

We next aimed to develop a glial model of ARSACS, deleting sacsin in C6 cells by means of a CRISPR/Cas9 approach (Figure 1E). We isolated 96 individual clones, of which 42% (40/96) survived and proliferated. Eighteen clones were probed for sacsin protein expression, and around 17% (3/18) of the clones did not express detectable levels of sacsin. We randomly selected one of them for further studies (Figure 1F).

### 3.2. Sacsin Loss Induces Higher Sensitivity to Oxidative Challenge

Mitochondrial alterations and oxidative stress are common hallmarks in neurodegenerative disorders, and ARSACS is no exception [7,21]. C6^Sacs−/−^ cells showed a significantly higher level of basal oxidative stress (Figure 2A). Next, reference C6 cells and the C6^Sacs−/−^ strain were treated with rotenone, a mitochondrial Complex I inhibitor, to determine whether sacsin loss could undermine their response to oxidative challenges. Incubation with the rotenone for 4 h had similarly mild toxicity in wild type and C6^Sacs−/−^ cells (Figure 2B,C; Appendix A). Neither MTT or LDH cytotoxicity assays gave indications of significant toxicity in these conditions in both cell lines (Figure 2B; Appendix A). However, a flow-cytometry-based DAPI exclusion assay suggested a non-significant tendency to higher toxicity in C6^Sacs−/−^ cells (Appendix A), and these cells showed a stronger decrease in cell size in C6^Sacs−/−^ cells (forward scatter, FSC-A, Figure 2D) consistent with a higher degree of damage. C6^Sacs−/−^ cells also show a significantly higher increase in oxidative stress upon rotenone exposure, often accompanied by a slight increase in DAPI staining, which is indicative of membrane damage (Figure 2E–I).

### 3.3. Sacsin Deletion Leads to Juxtanuclear Accumulation of Glial IFs

Sacsin deletion in neuronal cells induces the accumulation of neurofilament light, medium and heavy polypeptides (NFL, NFM and NFH, respectively), peripherin, α-internexin and vimentin in the juxtanuclear region, with the concomitant depletion of mitochondria in the same region [9,10]. Immunocytochemistry analysis unmasked a similar profile for vimentin, nestin and GFAP in C6^Sacs−/−^ cells (Figure 3A–C; Appendix A): juxtanuclear accumulation of the three glial intermediate filaments and depletion of mitochondria from this region. Transformation of widefield images by the Nano J Super-Resolution Radial Fluctuations (SRRF) ImageJ plugin [22] showed more condensed IF networks in this juxtanuclear region (Figure 3B). However, the impact was different for each IF: approximately 40% of cells showed accumulation of GFAP, 60 % of nestin, and 60 % of Vimentin (Figure 3C). The protein levels of glial IFs were generally higher in C6^Sacs−/−^ than in reference C6 cells (Figure 4A), but the increase in nestin levels did not achieve significance (Figure 4B). Filter trap assays supported the aggregation of these IFs (Figure 4C), although the increase in vimentin aggregation did not achieve statistical significance (Figure 4D). No significant alterations were observed in the organization of actin or microtubule networks (Appendix A). These data indicate that sacsin deficiency also disrupted the glial IF networks and induced mitochondrial network remodeling; the process occurs in neurons.

### 3.4. Sacsin Deletion Produces Alterations in the Response to Inflammatory Cytokines

Astroglia participates in neuroinflammation, where cytokine signaling plays a key role [23], and IFs are currently considered important scaffolds actively involved in signal transduction [24]. We hypothesized that sacsin deletion could interfere with specific signaling/inflammatory pathways, such as the Signal Transducer and Activator of Transcription 3 (STAT3) pathway, which play key roles in neuroinflammation [25,26]. Reference C6 cells responds poorly to LIF, IL-6 (Figure 5A) and Tumor Necrosis Factor alpha (data not shown). However, they show STAT3 activation after 20 min of incubation with IL-6 in combination with its soluble receptor IL-6R (Figure 5A–E), as determined by key STAT3 post-translational modifications, such as Y705 and S727 phosphorylation or K49 acetylation. This activation is accompanied by a significant increase in total STAT3 levels, which is not observed in C6^Sacs−/−^ cells (Figure 5B). The lower levels of total STAT3 C6^Sacs−/−^ cells could explain why we also observe lower levels of post-translational modifications in these cells, as the corresponding PTM/total STAT3 ratios are not significantly different from reference C6 cells (Figure 5C–E). These results suggest that the impairment in STAT3 signaling is not by direct regulation of STAT3 modifications, but by regulation of STAT3 levels, a hypothesis that is consistent with sacsin’s role as a chaperone.

## 4. Discussion

Sacsin mRNA is present in most tissues, although its expression levels show some cell specificity (source: The Protein Atlas). In the central nervous system, it was described to have higher levels in Purkinje cells, pyramidal neurons, thalamic and pontine nuclei and reticular formation [4]. However, public transcriptomics data indicated that sacsin is also expressed in glial cells, including astrocytes, Müller glia, oligodendrocyte precursor cells, mature oligodendrocytes and microglia [17,18]. Astrocytes express sacsin RNA levels as high as neurons, especially in younger individuals [17]. Our results confirm that the sacsin protein is present at high levels in astrocytes, and possibly in microglia, since we could detect sacsin in N9 microglial cells.

To the best of our knowledge, scientific studies of sacsin function focused almost exclusively on neurons, and a possible role of glial cells in ARSACS pathogenesis is completely unknown. We developed a new cellular model to investigate the role of sacsin in glial cells based on C6 rat glioma cells (Figure 1). The fact that they are rodent cells will enable comparative studies in mice models of ARSACS and primary astroglial cells from rats. The loss of sacsin in C6 cells produces the accumulation of intermediate filaments—vimentin, nestin and GFAP—in the juxtanuclear area, where there is a concomitant depletion of mitochondria (Figure 3 and Figure 4). These results are consistent with previous reports where sacsin knockout from human HEK-293T (embryonic kidney) and SH-SY5Y (neuroblastoma) cells induced alterations in vimentin and/or neurofilament networks [9,10]. Sacsin is also expressed in keratinocytes and fibroblasts, and ARSACS patients showed skin alterations and lipofuscin deposits [3]. These data suggest that sacsin could have a far-reaching role in the organization and dynamics of different intermediate filaments beyond neurons, and ARSACS could belong to the growing family of IF-pathies, as AxD or GAN [13,15,27].

Although ARSACS, AxD and GAN are different genetic diseases with their own clinical profiles [2,15,28,29], the similarities between some of their symptoms (e.g., pediatric diagnosis, ataxia, dysarthria, nystagmus) and histopathological features (e.g., white matter loss, IF disruption and, at least in ARSACS and GAN mitochondrial dysmotility) are remarkable [4,27,30,31]. The three pathologies have mixed features of neurodegenerative and neurodevelopmental disorders. Vimentin, nestin and GFAP are also typically found in neural precursor cells [32,33], and C6 cells show properties of both astroglia and neural precursor cells [34,35]. Although we were unable to detect sacsin in adult neural precursor cells from the rat dentate gyrus and the subventricular zone (Data not shown), sacsin could regulate the organization of IFs in precursor cells during development, with potential implications for the disease onset. Our results should encourage further studies in these directions.

Oxidative stress and neuroinflammation are central to neurodegenerative diseases, and astrocytes are major players in these processes. IFs are involved in signal transduction as scaffolds for signaling proteins and mitochondrial motility [24,36], and disruption of GFAP alone in AxD models disrupts various signaling pathways [37]. Our results indicate that sacsin deletion impairs the REDOX status of C6 cells and their response of cells to oxidative challenges and cytokines (Figure 2 and Figure 5). The impairment of STAT3 signaling in C6^Sacs−/−^ cells seems to be mediated by the regulation of STAT3 levels rather than its post-translational modifications (Figure 5). Bearing in mind the role of sacsin as a chaperone [5,6], it is possible that it contributes directly or indirectly to STAT3 folding or stability. The consequences of disrupting the response of glia to inflammatory cues are difficult to predict, but could be relevant for disease onset and progression, especially if this disruption affects both astroglia and microglia.

In summary, to the best of our knowledge, our study is the first to show that astrocytes express sacsin at the protein level and that their depletion in glial-like cells causes pathological hallmarks of ARSACS similar to those observed in neuronal cells. Sacsin knockout cells showed a dysregulation in their REDOX balance and altered responses to inflammatory cues. These data support a potential role of astroglia, microglia or even neural precursor cells in ARSACS, which should be further analyzed. Future studies should test our findings in postmortem brain tissues from ARSACS patients and existing mouse models of the disease, and analyze how the functions of primary astroglia are disrupted by sacsin loss. Considering the developmental nature of ARSACS, these studies should probably focus on embryonic or perinatal stages.

## Figures and Tables

**Figure 1 cells-11-00299-f001:**
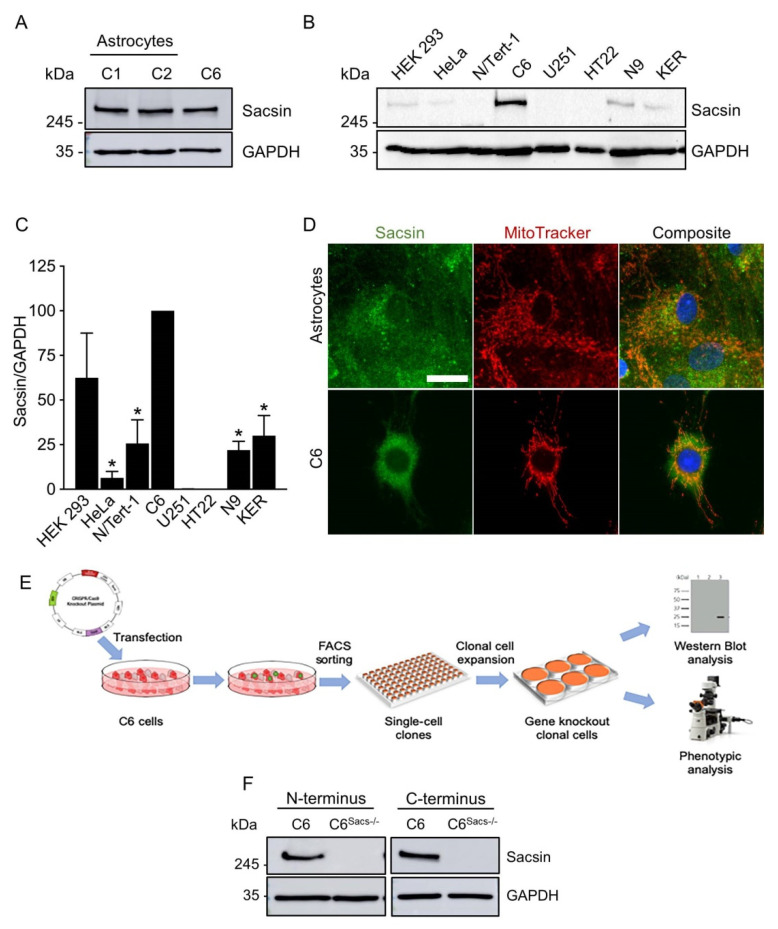
Glial cells express sacsin. (**A**) Representative Western blot images indicate that primary rat astrocytes have sacsin levels similar to C6 glioblastoma cells, with C1 and C2 indicating independent cultures of primary astrocytes. (**B**) Sacsin was detected in HEK293, HeLa, N/Tert-1 and N9 cell lines and primary human keratinocytes by Western blot, but at lower levels than C6 cells. (**C**) Sacsin was not detected in HT22 mouse hippocampal cell lines or U251 glioblastoma cells in these conditions, but we cannot rule out that they express very low levels of the protein. Data were analyzed by means of a one-way ANOVA, followed by a Tukey post hoc test, *, significant vs. C6 reference strain, *p* < 0.05. (**D**) Representative images of immunocytochemistry for endogenous sacsin (green) in primary rat astrocytes and C6 cells. Mitochondria were counterstained with Mitotracker (red) and nuclei with Hoechst (blue). Scale bar, 20 μm. (**E**), Diagram illustrating the workflow for CRISPR/Cas9-mediated generation of C6^Sacs−/−^ cell lines. Cells were transfected with CRISPR/Cas9 plasmids, sorted by FACS and clonally expanded. Resulting clonal populations were then tested for sacsin expression and phenotype. (**F**) Representative Western blot analyses of a C6^Sacs−/−^ clone, using two different sacsin antibodies against its N- and C-termini (refs. sc-515118 and ABN1019, respectively).

**Figure 2 cells-11-00299-f002:**
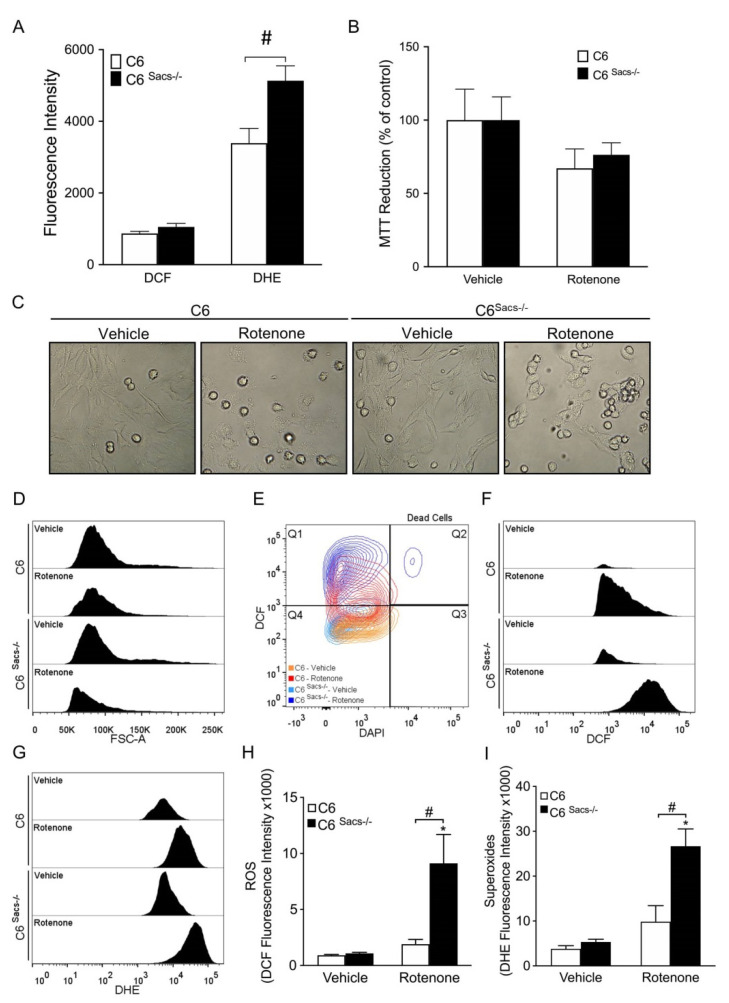
Sacsin knockout renders cells more sensitive to rotenone-induced stress. (**A**) Under basal conditions, C6^Sacs−/−^ had levels of reactive oxygen species (ROS) similar to C6 cells, as determined by DCF fluorescence. However, C6^Sacs−/−^ showed higher levels of superoxide radicals, as determined by DHE fluorescence. Data were analyzed by means of two-way ANOVA followed by a Tukey post hoc test, #, significant vs. C6 reference strain, *p* < 0.05. (**B**) Both C6 and C6^Sacs−/−^ displayed a non-significant decrease in viability when challenged with rotenone (5 µM) or its vehicle (DMSO 0.1%) for 4 h. (**C**) Representative brightfield images show similar gross morphological alterations in both cell strains following rotenone treatment, but flow cytometry analysis (**D**) indicated a stronger decrease in size (Forward-Scatter, FSC-A) in C6^Sacs−/−^ cells. (**E**) Representative flow cytometry plot showing DAPI and DCF staining of C6 and C6^Sacs−/−^ cells after incubation with rotenone (red and dark blue, respectively) or its vehicle (orange and light blue). Rotenone increases ROS levels in both C6 cell strains, but with more intensity in C6^Sacs−/−^ (Quadrant 1, Q1). In C6^Sacs−/−^ cells it only induces some residual cell death (DAPI staining, quadrants 2 and 3, Q2/Q3). (**F**,**G**) Representative histograms from flow cytometry analysis using DCF and DHE in C6 and C6^Sacs−/−^ cells after treatment with rotenone. (**H**,**I**) Quantification of oxidative stress levels in 3 independent experiments (mean ± SEM). Data were analyzed by means of two-way ANOVA, followed by a Tukey post hoc test, *, significant vs. vehicle; #, significant vs. C6 reference strain, *p* < 0.05.

**Figure 3 cells-11-00299-f003:**
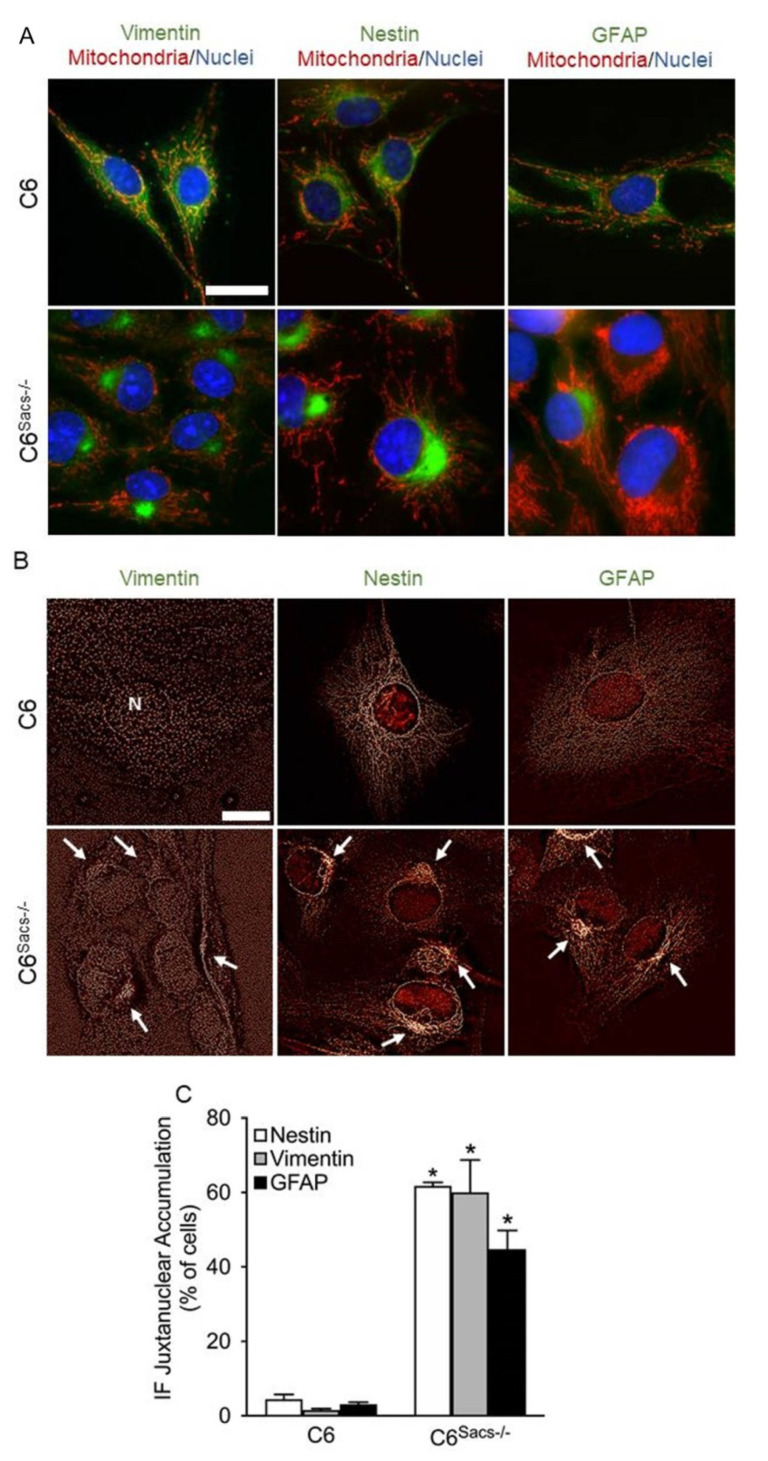
Sacsin deletion disrupts glial intermediate filament networks. (**A**) Representative immunocytochemistry images showing the distribution of the glial intermediate filaments vimentin, nestin and GFAP (green); mitochondria (Mitotracker, red); and nuclei (Hoechst, blue) in C6 and C6^Sacs−/−^ cells. Vimentin, nestin and GFAP accumulate in the juxtanuclear area in C6^Sacs−/−^ cells. Scale bar, 20 μm. (**B**) Widefield images of C6 and C6^Sacs−/−^ cells were further analyzed by means of the Nano J Super-Resolution Radial Fluctuations (SRRF) algorithm, which provided higher resolution details to obtain a more defined image of the intermediate filament networks. N, nucleus. White arrows, intermediate filament aggregates. Scale bar, 20 μm. (**C**) Quantification of microscopy images from 3 independent experiments (mean ± SEM). *, significant vs. C6 reference strain (*p* < 0.05, Student’s *t*-test). The total numbers of reference C6 cells counted in 3 independent experiments were 549 (GFAP), 464 (Nestin) and 353 (Vimentin). The total numbers of C6^Sacs−/−^ cells counted in 3 independent experiments were 836 (GFAP), 697 (Nestin) and 438 (Vimentin).

**Figure 4 cells-11-00299-f004:**
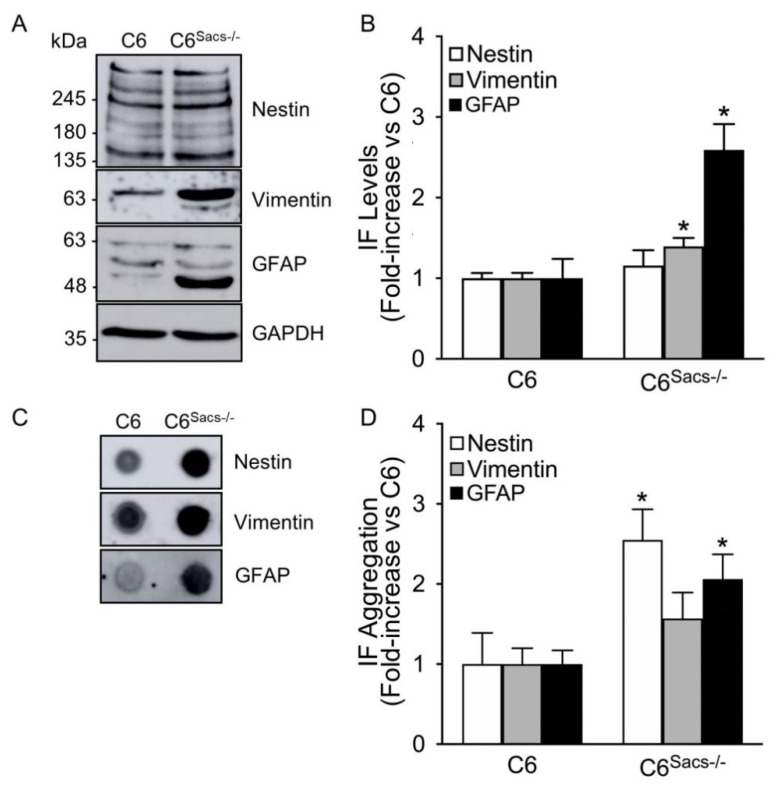
Sacsin deletion increases protein levels and aggregation of glial intermediate filaments. (**A**) Representative Western blots showing the expression patterns of the glial intermediate filaments. (**B**) Densitometric analysis of Western blots from at least 3 independent experiments normalized versus GAPDH. The levels of GFAP and vimentin proteins are increased in C6^Sacs−/−^cells. (**C**) Filter trap assays confirmed higher aggregation of intermediate filaments in C6^Sacs−/−^cells. (**D**) Densitometric analysis of filter traps normalized versus the reference C6 strain. *, significant vs. C6 reference strain (*p* < 0.05, Student’s *t*-test).

**Figure 5 cells-11-00299-f005:**
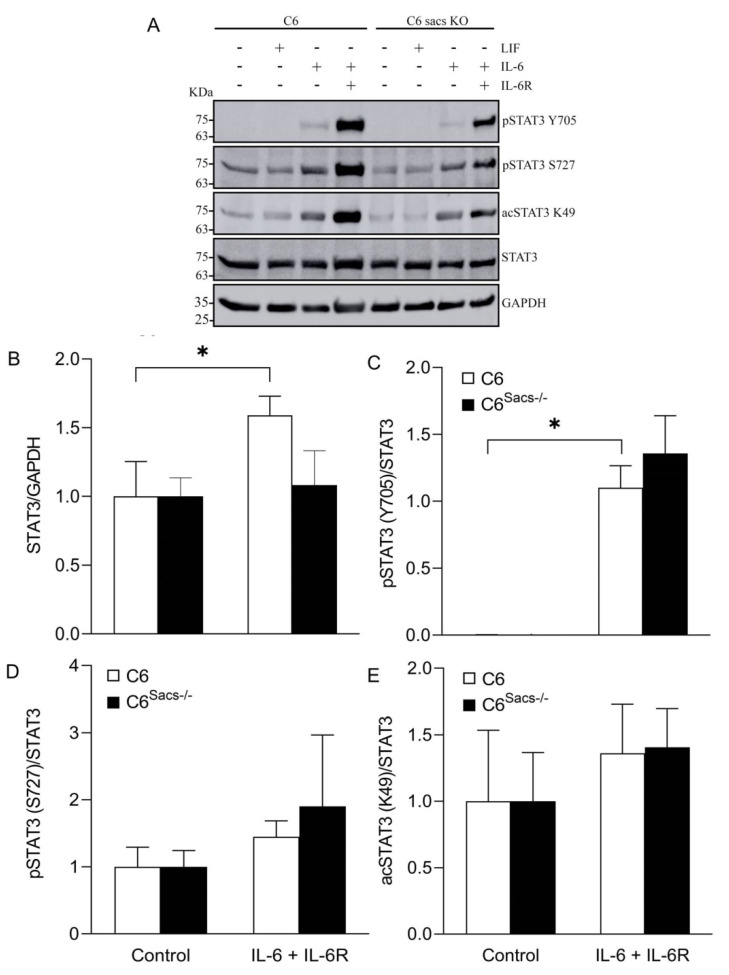
Sacsin deletion impairs response to inflammatory cytokines. Reference C6 and C6^Sacs−/−^ strains were incubated with LIF and IL-6 cytokines (200 and 30 ng/mL, respectively) for 20 min, but a strong response was only observed upon the addition of the soluble IL-6 receptor (IL-6R, 60 ng/mL) in combination with IL-6. Rate-limiting, post-translational modifications of STAT3 were used as surrogates of STAT3 activation, namely Y705 and S727 phosphorylation and K49 acetylation. (**A**) Representative Western blot images. (**B**–**E**) Densitometry analysis of bands from at least 3 independent experiments normalized to GAPDH or total STAT3. White bars, reference C6 cell strain; black bars, C6^Sacs−/−^ cells. Data were analyzed by means of two-way ANOVA, followed by a Tukey post hoc test. * *p* < 0.05, significant vs. IL-6+IL-6R in C6 cell strain.

## Data Availability

Not applicable.

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
