# Peer review of "Sacsin Deletion Induces Aggregation of Glial Intermediate Filaments"

_cells, 2022, doi:10.3390/cells11020299_

Round 1

Reviewer 1 Report

In their study, Murtinheira et al. demonstrated for the first time that deletion of the protein sacsin in astrocytes causes a set of key pathological features that have been reported in other cell types where sacsin function is altered or absent such as that found in Autosomal recessive spastic ataxia of Charlevoix-Saguenay (ARSACS). ARSACS is a rare and poorly understood genetic neurodegenerative disorder with onset as early as 1 year of age. Therefore, any studies on the pathological mechanisms underlying ARSACS are extremely valuable. Given the role that astrocytes play in regulating brain physiology, neuronal function and survival, the findings by the authors are a significant contribution that shed light onto the role of astrocytes in ARSACS and lay ground for future important experiments. These will be made easy by the cell line model of sacsin knockout in C6 rat glioma cells that was newly developed by the authors. Overall, the manuscript is exceptionally well written and I think it deserves publication. Statistics were applied correctly.

Main comments:

  1. In addition to the Human Protein Atlas, the authors should check the https://www.brainrnaseq.org (from the Barres lab) for gene expression on a brain cell type-basis for both mice and humans. They would note that the expression of sacs, albeit modest, seems to be the highest in astrocytes and neurons. This information can be added to back up their premise.

  1. The authors need to explain why they used anti-GP130 (glial marker). This could just be added to the Figure 1 legend.

  1. How was the fluorescence of DCF and DHE in Figure 2A determined? (by plate reader?) It should differ from those in Figures 2H-I determined by flow cytometry.

  1. The MTT assay can be tricky to assess cell survival. Did the authors measure the LDH release upon treating cells with rotenone? The DCF data were very clear. Alternatively, perhaps if they try to plot the intensity of DAPI alone that could be informative. Looking at Figure 2E it does look like that DAPI increases to a larger extent in the Sacs-/- cells exposed to rotenone relative to vehicle.

  1. The mitochondria of Sacs-/- cells in Figure 3A appear to be more elongated. If this is the case, perhaps it is worth highlighting that.

  1. Why is that the distribution pattern of Vimentin in Figure 3B looks distinct from Nestin and GFAP when in Figure 3A they seem to be alike, for both wildtype and Sacs-/-?

  1. “No significant alterations were observed in the organization of actin or microtubule networks (Data not shown).” If the authors have these pictures at hand (one of them or both), it would be interesting to see how the rest of the cytoskeleton looks like in wildtype and Sacs-/- cells.

  1. The Discussion could be improved by adding/addressing the following:

Future studies should focus on testing and confirming the findings presented by the authors (and others) in post-mortem brain tissue from ARSACS patients as well as in the mouse model of sacsin knockout and the new cell line generated.

In addition, it should be tested how the knockout of sacsin compares to the mutations found in human patients, specifically whether these mutations lead to a full loss of function of the protein or some other situation.

Finally, they should add that the role of ARSACS pathology in astrocytes that they have unveiled through the knockout of sacsin should be further investigated by addressing in depth how the different astrocytic functions are disrupted, namely those associated with astrocyte-neuron interactions (for example, via the glutamine-glutamate cycle). Eventually, these studies should be validated using primary astrocytes.

Minor comments/typos:

Line 54. “… from the plakin…”

Lin 55. “… IFs comprise an extremely diverse family of proteins…”

Line 148. How exactly were the brains dissociated? Trypsin for 30 min at 37°C…?

Line 355. “… far-reaching…”

Author Response

We would like to sincerely thank this reviewer for his kind and insightful revision of our manuscript. We believe we have addressed most of his concerns, and the manuscript has improved considerably. Please, find a point-by-point reply to the issues raised by the reviewer. We hope you find the revised version of our article suitable for publication in Cells.

Reviewer 1

1. In addition to the Human Protein Atlas, the authors should check the https://www.brainrnaseq.org (from the Barres lab) for gene expression on a brain cell type-basis for both mice and humans. They would note that the expression of sacs, albeit modest, seems to be the highest in astrocytes and neurons. This information can be added to back up their premise.

Response: Thank you very much for your excellent suggestion, which really enriched and reinforced our premises. The BrainRNAseq database not only showed that astrocytes expressed levels of RNA as high as neurons in both mouse and human, but also that astrocytes expressed higher levels in younger individuals, which is consistent with the fact that ARSACS is manifested in infants. We have integrated this information in the manuscript with the corresponding references.

2. The authors need to explain why they used anti-GP130 (glial marker). This could just be added to the Figure 1 legend

Response: We used anti-GP130 as a loading control in the first western blots because it was expressed in both astroglia and C6 cells at similar levels and it was retained in the same weight range as sacsin. However, for the sake of consistency throughout the manuscript, we have changed these images by a western blot of the samples using GAPDH as a loading control.

3. How was the fluorescence of DCF and DHE in Figure 2A determined? (by plate reader?) It should differ from those in Figures 2H-I determined by flow cytometry.

Response: The fluorescence of DCF and DHE in Figure 2A was determined by means of flow cytometry. We did not use a plate reader to determine DCF, DHE or DAPI. The difference between these data and data represented in Figures 2H and 2I is that in Figure 2A there is no vehicle (DMSO) added to cells. Statistics also changed between the two situations. In Figure 2A, we used a t-student, as there were only two experimental conditions (reference C6 cells vs C6sacs-/- cells). In Figures 2H-I, we used a Two-way ANOVA as indicated in the legend, because there were 4 experimental conditions (each cell strain with or without rotenone). The missing information is now included in the legend for clarity.

4. The MTT assay can be tricky to assess cell survival. Did the authors measure the LDH release upon treating cells with rotenone? The DCF data were very clear. Alternatively, perhaps if they try to plot the intensity of DAPI alone that could be informative. Looking at Figure 2E it does look like that DAPI increases to a larger extent in the Sacs-/- cells exposed to rotenone relative to vehicle.

Response: We did both LDH release and DAPI quantification with similar results. These data are now included as Supplementary Figure 1, and commented in the Results section. DAPI and FSC data suggested higher damage in C6Sacs-/- cells, but did not achieve statistical significance, because there is high variability between experiments. Rotenone treatment is only 4 hours, and there is little time for cell death to occur at significant levels, but we observe a clear increase in ROS consistent with early alterations. Longer periods (16 hours) were also tried, but produced massive toxicity, making it difficult to ascertain whether sacsin -/- cells were really more sensitive than reference C6 cells at that advanced toxicity stage.

5. The mitochondria of Sacs-/- cells in Figure 3A appear to be more elongated. If this is the case, perhaps it is worth highlighting that.

Response: We have observed gross alterations in mitochondrial networks, but were unable to analyze them either qualitatively or quantitatively. Qualitatively, it is not clear if mitochondria are elongated or overcrowding the space left for them out of the IF aggregates; if there is a reduction in fission, an increase in fusion, or they are simply overlapping; etc…. Quantitatively, we are still trying to master the available tools, but have not made enough progress to publish a rigorous analysis. In summary, we had to decide whether making a speculative statement that we could only back up with examples of images or avoiding any statement at all. We believe our images already hint to alterations, but we would prefer not to speculate.

6. Why is that the distribution pattern of Vimentin in Figure 3B looks distinct from Nestin and GFAP when in Figure 3A they seem to be alike, for both wildtype and Sacs-/-?

Response: The SRRF plugin works on 100 images taken from the same cells in a very short time. The idea is that specific fluorescence will be present through many of the pictures while non-specific fluorescence will be less stable, more transitory. Then, the plugin integrates them into a single picture, highlighting the most stable fluorescence signals. The plugin is extremely sensitive to background fluorescence, and can register it even when it is not detected by the bare eye. In this sense, the vimentin antibody is dirtier than the others, producing background signals that are not intense enough to be clearly visible in a single widefield picture (only increasing substantially the brightness), but detected by the SRRF plugin in 100 pictures. For this reason, the vimentin filaments are less differentiated from the nuclei and background. However, the morphological changes of vimentin are still similar to those observed for GFAP and Nestin. Additionally, we corrected a mistake in the scale in that picture, which was smaller than it should, and we think now it is clearer.  

7. “No significant alterations were observed in the organization of actin or microtubule networks (Data not shown).” If the authors have these pictures at hand (one of them or both), it would be interesting to see how the rest of the cytoskeleton looks like in wildtype and Sacs-/- cells.

Response: We agree and apologize for not including them in the first place. We have included a few representative images as Supplementary Figure 3.

8. The Discussion could be improved by adding/addressing the following:

Future studies should focus on testing and confirming the findings presented by the authors (and others) in post-mortem brain tissue from ARSACS patients as well as in the mouse model of sacsin knockout and the new cell line generated.

In addition, it should be tested how the knockout of sacsin compares to the mutations found in human patients, specifically whether these mutations lead to a full loss of function of the protein or some other situation.

Finally, they should add that the role of ARSACS pathology in astrocytes that they have unveiled through the knockout of sacsin should be further investigated by addressing in depth how the different astrocytic functions are disrupted, namely those associated with astrocyte-neuron interactions (for example, via the glutamine-glutamate cycle). Eventually, these studies should be validated using primary astrocytes.

Response: We agree and have introduced these subjects in the end of the discussion.

9. Minor comments/typos:

Line 54. “… from the plakin…”

Lin 55. “… IFs comprise an extremely diverse family of proteins…”

Line 148. How exactly were the brains dissociated? Trypsin for 30 min at 37°C…?

Line 355. “… far-reaching…”

Response: These issues are now corrected in the revised version. Thank you very much for reading our manuscript to this level of detail.

Reviewer 2 Report

The article by Murtinheira et al., entitled “Sacsin deletion induces aggregation of glial intermediate filaments”, showed that the sacsin protein is expressed in astroglia. The authors developed an astroglial model of ARSACS by deleting sacsin in the C6 rat glioma cell line. Their results indicate that sacsin also regulates glial intermediate filaments organization, and suggest a potential link between ARSACS and other neurodegenerative conditions diagnosed in infants, such as Alexander disease (AxD) or Giant Axonal Neuropathy (GAN). In particular, sacsin knockout in C6 cells (C6Sacs-/-) induced accumulation of the glial intermediate filaments glial fibrillary acidic protein (GFAP), nestin and vimentin in the juxtanuclear area, and a concomitant depletion of mitochondria. Moreover, C6Sacs-/- cells showed impaired responses to oxidative challenges (Rotenone) and inflammatory stimuli (Inter-leukin-6).

Although it is original, developing an astroglial model of ARSACS by deleting sacsin in the C6 rat glioma cell line, well designed and expanded there are some methodological points to review:

- Generation of sacsin KO cell lines using the CRISPR/Cas9 system.

To verify that a protein of interest has been ablated is demonstrating its absence by western blot analysis. However, single western blots may be misleading, and characterizing the precise alterations caused by CRISPR at the DNA level may provide additional useful information (i.e. homo- or heterozygous cell clones). Thus, I would suggest to the authors to verified the correct gene editing (knockout), in the selected clone, also by Sanger sequencing or a PCR-based screening strategy.

- Figure 1E. Representative western blot analyses of a C6Sacs-/- clone, using two different sacsin antibodies against its N- and C-termini.

  1. Why didn't the authors choose to run the samples on a gel with a lower percentage of polyacrylamide in order to better separate the bands relative to sacsin protein and uniquely identify them? Which band did they choose?
  2. The authors state to use two sacsin antibodies against its N-terminal and C-terminal. Please, clearly specify the company for each one (N- or C-terminal) in the M&M section.

- Figure 3D. Representative western blots showing the expression patterns of the glial intermediate filaments. The levels of GFAP and vimentin proteins are increased in C6Sacs-/-cells.

The authors cannot comment on the expression levels of studied proteins without analyze their relative expression levels, normalized to GAPDH, from 3 replicate experiments. Please, add the densitometry ratio for each protein performing the corresponding statistical analysis (mean of the experiments and standard deviation).

- Figure 3E. Filter trap assays confirmed higher aggregation of intermediate filaments in C6Sacs-/- cells.

  1. Please, add if possible a quantitative graph representing the relative aggregation amount for each protein in C6Sacs-/- cells compare to C6 ones.
  2. Please, describe the filter trap assay in the Materials and Methods section.

Author Response

We would like to sincerely thank this reviewer for his kind and insightful revision of our manuscript. We believe we have addressed most of his concerns, and the manuscript has improved considerably. Please, find a point-by-point reply to the issues raised by the reviewer. We hope you find the revised version of our article suitable for publication in Cells.

Reviewer 2

1. Generation of sacsin KO cell lines using the CRISPR/Cas9 system.

To verify that a protein of interest has been ablated is demonstrating its absence by western blot analysis. However, single western blots may be misleading, and characterizing the precise alterations caused by CRISPR at the DNA level may provide additional useful information (i.e. homo- or heterozygous cell clones). Thus, I would suggest to the authors to verify the correct gene editing (knockout), in the selected clone, also by Sanger sequencing or a PCR-based screening strategy.

Response: We agree that this information is not essential but could be very useful, and we actually tried. However, we have been unable to verify the gene edition by a PCR-based strategy with 3 different sets of primers designed using the consensus mouse sequence for sacsin, and using genomic DNA from both C6 strains as template. We even tried (by mistake and also unsuccessfully) 3 pairs of primers using the human consensus sequence for the gene. Although we were able to amplify a few fragments, they were never from sacsin. At this point, we are not sure what the issue is, but we are still working on it. However, we believe we have already made the point that these cells have lost sacsin expression. This should be enough to support a causal relationship between the phenotype we observe and the absence of this protein, which was the main goal of this report.

2. Figure 1E. Representative western blot analyses of a C6Sacs-/- clone, using two different sacsin antibodies against its N- and C-termini.

2a. Why didn't the authors choose to run the samples on a gel with a lower percentage of polyacrylamide in order to better separate the bands relative to sacsin protein and uniquely identify them? Which band did they choose?

Response: The percentage of acrylamide is generally fine but, for unknown reasons, sometimes sacsin antibodies show several bands or a smear in a weight range compatible with sacsin molecular weight. These bands or smears do not occur always in reference C6 cells and are never present in sacs -/- C6 cells. Thus, we disregarded them as either alternative splicing or, more likely, proteolytic fragments. In order to avoid confusions, we have changed these western blots by others where sacsin bands are sharper.

2b. The authors state to use two sacsin antibodies against its N-terminal and C-terminal. Please, clearly specify the company for each one (N- or C-terminal) in the M&M section.

Response: Thank you very much for identifying this missing information. It is now included in the Material and Methods section, as requested.

3. Figure 3D. Representative western blots showing the expression patterns of the glial intermediate filaments. The levels of GFAP and vimentin proteins are increased in C6Sacs-/-cells.

The authors cannot comment on the expression levels of studied proteins without analyze their relative expression levels, normalized to GAPDH, from 3 replicate experiments. Please, add the densitometry ratio for each protein performing the corresponding statistical analysis (mean of the experiments and standard deviation).

Response: We have added the densitometry of western blots and analyzed it statistically, as requested.

4. Figure 3E. Filter trap assays confirmed higher aggregation of intermediate filaments in C6Sacs-/- cells.

4a. Please, add if possible a quantitative graph representing the relative aggregation amount for each protein in C6Sacs-/- cells compare to C6 ones.

4b. Please, describe the filter trap assay in the Materials and Methods section.

Response: We have added the densitometry of filter traps and analyzed it statistically, and added the Filter Trap Assay in the Material and Methods, as requested. Thank you very much for identifying this missing information, which was essential to reproduce our results.

Reviewer 3 Report

In this manuscript, the authors developed a sacsin (SACS) knockout C6 cell (rat glioma) model to study the role of sacsin in intermediate filament organization in glia cells. They found that sacsin depletion leads to the accumulation of GFAP, nestin and vimentin around the nucleus, and a concomitant depletion of mitochondria in that area. Their results also suggest that knockout cells are more sensitive to rotenone-induced stress.

The manuscript is well written and provides relevant molecular findings for ARSACS pathology. However, there are some results that need to be clarified, and the discussion section can be improved. Thus, I suggest that authors provide further details and revise some issues as described below.

  1. Line 243-244 (Figure 1): “Sacsin was not detected in HT22 mouse hippocampal cell line or U251 glioblastoma cells”. Despite not being visible in the blot, sacsin can present at low levels in those cells. The authors should mention this assumption. Moreover, as the GAPDH bands are juxtaposed, it would be more informative and clearer if the authors provide a quantitative graph of the sacsin levels in all cell lines.
  1. Line 268-269 (Figure 2): The presence of one asterisk or two asterisks in the figure give the impression that the values of p are <0.05 or <0.01. Can the authors clarify this?
  2. Line 319 (Figure 3): On figure A can the authors present the fluorescence images separately for each channel/protein? It will help to confirm the depletion of mitochondria in the area where intermediate filaments accumulate. On figure D, quantitative data of protein levels should be shown.
  3. Line 334 (Figure 4): The levels of phosphorylated and acetylated STAT3 should be normalized to total STAT3 levels - to ensure that differences are indeed due to post translational modifications on STAT3 and not due to differences in STAT3 levels.

Author Response

We would like to sincerely thank this reviewer for his kind and insightful revision of our manuscript. We believe we have addressed most of his concerns, and the manuscript has improved considerably. Please, find below a point-by-point reply to the issues raised by the reviewer. We hope you find the revised version of our article suitable for publication in Cells.

Reviewer 3

1. Line 243-244 (Figure 1): “Sacsin was not detected in HT22 mouse hippocampal cell line or U251 glioblastoma cells”. Despite not being visible in the blot, sacsin can present at low levels in those cells. The authors should mention this assumption. Moreover, as the GAPDH bands are juxtaposed, it would be more informative and clearer if the authors provide a quantitative graph of the sacsin levels in all cell lines.

Response: We fully agree with the reviewer and apologize for the assumption. We have included it in the legend of Figure 1, and added the densitometry of western blots with the corresponding statistical analysis.

2. Line 268-269 (Figure 2): The presence of one asterisk or two asterisks in the figure give the impression that the values of p are <0.05 or <0.01. Can the authors clarify this?

Response: We apologize for the confusion. The double asterisk is indeed used most frequently to indicate p<0.01, and we wanted to indicate comparisons versus different experimental groups (vehicle-treated cells or the reference C6 strain). We have now changed the double asterisk by the cardinal symbol (#) in Figure 2. We believe this is now much clearer for the reader.

3. Line 319 (Figure 3): On figure A can the authors present the fluorescence images separately for each channel/protein? It will help to confirm the depletion of mitochondria in the area where intermediate filaments accumulate. On figure D, quantitative data of protein levels should be shown.

Response: We agree, and we have included the images for each channel as Supplementary Figure 2, as requested. We have also added the densitometry analysis to all western blots and filter traps of the manuscript.

4. Line 334 (Figure 4): The levels of phosphorylated and acetylated STAT3 should be normalized to total STAT3 levels - to ensure that differences are indeed due to post translational modifications on STAT3 and not due to differences in STAT3 levels.

Response: We agree and apologize for this mistake, as we recognize that this is the correct way to present these sorts of data. We have normalized STAT3 PTMs versus total STAT3, and the results have indeed changed substantially. It seems that sacsin loss disrupts STAT3 signaling by means of the control of STAT3 levels, rather than its post-translational modifications. This is consistent with a role for sacsin as a chaperone, and does not change substantially our conclusions. If there is less STAT3, and as a consequence there is less activated STAT3, as we observe, this can have a negative effect in the transduction of inflammation signals. This is now described in the main text in both the Results and Discussion sections.

Round 2

Reviewer 3 Report

The authors have satisfactorily addressed the comments raised in the previous round of review.